# Planar thermal Hall effect from phonons in a Kitaev candidate material

Lu Chen [1,4] ✉, Étienne Lefrançois [1,4], Ashvini Vallipuram[1], Quentin Barthélemy[1], Amirreza Ataei[1], Weiliang Yao[2], Yuan Li [2] & Louis Taillefer [1,3] ✉

The thermal Hall effect has emerged as a potential probe of exotic excitations in spin liquids. In the Kitaev magnet $\alpha$-RuCl$_3$, the thermal Hall conductivity $\kappa_{xy}$ has been attributed to Majorana fermions, chiral magnons, or phonons. Theoretically, the former two types of heat carriers can generate a "planar" $\kappa_{xy}$, whereby the magnetic field is parallel to the heat current, but it is unknown whether phonons also could. Here we show that a planar $\kappa_{xy}$ is present in another Kitaev candidate material, Na$_2$Co$_2$TeO$_6$. Based on the striking similarity between $\kappa_{xy}$ and the phonon-dominated thermal conductivity $\kappa_{xx}$, we attribute the effect to phonons. We observe a large difference in $\kappa_{xy}$ between different configurations of heat current and magnetic field, which reveals that the direction of heat current matters in determining the planar $\kappa_{xy}$. Our observation calls for a re-evaluation of the planar $\kappa_{xy}$ observed in $\alpha$-RuCl$_3$.

The quest for quantum spin liquids (QSLs) has attracted tremendous interest due to the potential realization of non-Abelian statistics and novel exotic excitations[1]. A promising platform for the realization of QSLs is the Kitaev model, which features bond-dependent Ising interactions between spin-1/2 degrees of freedom on a honeycomb lattice[2]. The Kitaev model is exactly solvable, and it predicts the existence of itinerant Majorana fermions that carry heat and should therefore contribute to thermal transport[3]. A topologically protected edge current can emerge from the bulk Majorana bands under an external magnetic field and be detected by the thermal Hall effect as a half-quantized thermal Hall conductivity $\kappa_{xy}$[4,5].

The search for Kitaev QSLs in real materials has focused on 5$d$ iridium[6] and 4$d$ ruthenium compounds[7], of which the quasi-2D Mott insulator $\alpha$-RuCl$_3$ has been the most intensively studied. In $\alpha$-RuCl$_3$, antiferromagnetic (AF) order sets in below a temperature $T_N \simeq 7$ K, with a spin configuration called "zigzag" order, but the application of a magnetic field $H$ parallel to the honeycomb layers suppresses this order for $H \gtrsim 7$ T, thereby raising the possibility of a field-induced QSL state at low temperature when $H \gtrsim 7$ T. A half-quantized $\kappa_{xy}$ (i.e., $\kappa_{xy}^{2D}/T = \pi k_B^2/12\hbar$) was reported in $\alpha$-RuCl$_3$ – for an in-plane field in excess of 7 T – and interpreted as evidence of itinerant Majorana

fermions[8,9]. The half-quantized $\kappa_{xy}$ plateau appears even for a "planar" Hall configuration[10], i.e., when the magnetic field is applied within the 2D plane and parallel to the heat current $J$, specifically for $H // a$, where $a$ is the crystal direction perpendicular to the Ru-Ru bond (the so-called zigzag direction). Subsequently, Czajka et al. reported that the planar $\kappa_{xy}$ in $\alpha$-RuCl$_3$ shows no sign of half-quantization, and they instead attributed its smooth growth with temperature for $H // J // a$ to chiral magnons[11]. Theoretical work has shown that Majorana fermions[3] and topological magnons[12,13] are both able to generate a planar $\kappa_{xy}$ in $\alpha$-RuCl$_3$, when $H // a$.

In contrast to these two scenarios of exotic topological excitations, it has also been argued that phonons are the main carriers responsible for the thermal Hall effect in $\alpha$-RuCl$_3$ – at least for a field normal to the 2D planes ($H // c$ and $J // a$)[14]. The argument is based on the striking similarity of $\kappa_{xy}(T)$ to $\kappa_{xx}(T)$, the phonon-dominated longitudinal thermal conductivity. However, it remains unknown whether phonons can also generate a planar $\kappa_{xy}$, where $H // J // a$.

Note that a non-zero planar Hall effect – i.e. a non-zero $\Delta T_y$ for $H // x$ in Fig. 1c and d – is in principle only allowed if the crystal structure of a material breaks three symmetries: the $xy$ and $yz$ planes are *not* mirror planes, and the $C_2$ rotational symmetry is broken along

[1]Institut quantique, Département de physique & RQMP, Université de Sherbrooke, Sherbrooke, QC, Canada. [2]International Center for Quantum Materials, School of Physics, Peking University, Beijing, China. [3]Canadian Institute for Advanced Research, Toronto, ON, Canada. [4]These authors contributed equally: Lu Chen, Étienne Lefrançois. ✉e-mail: lu.chen@usherbrooke.ca; louis.taillefer@usherbrooke.ca

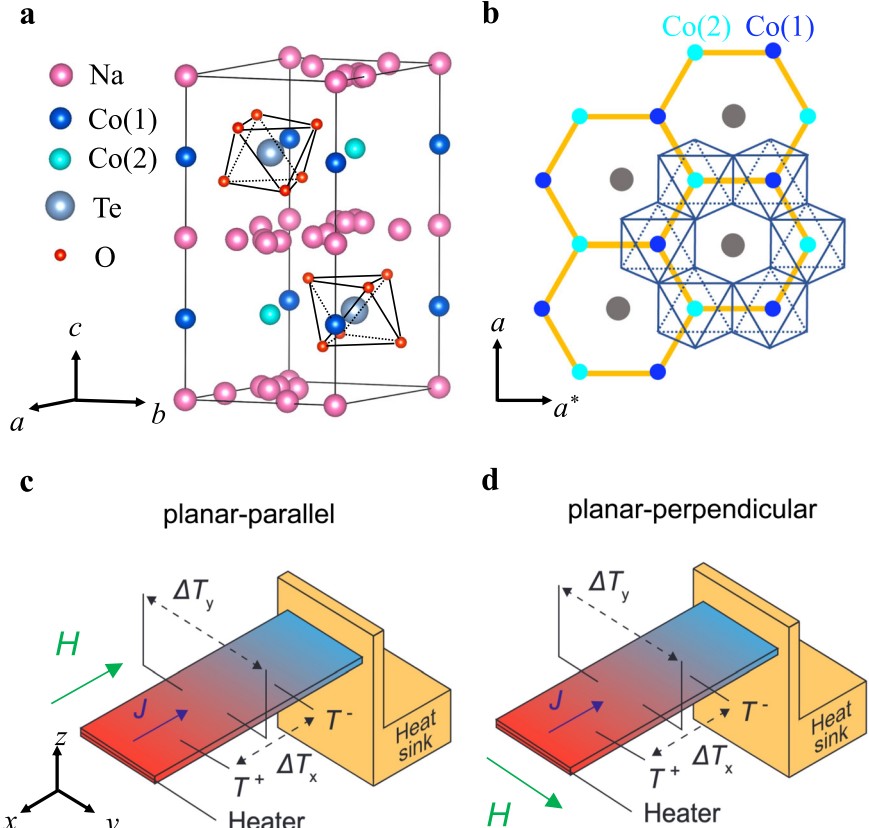

**Fig. 1 | Crystal structure of Na₂Co₂TeO₆ and experimental setup. a** The crystal structure of Na₂Co₂TeO₆. Honeycomb layers of edge-sharing CoO₆ octahedra sandwiched between Na layers and stacked along the $c$ direction in an *ABAB* format. The two types of inequivalent environments result in two different Co²⁺ sites which are labeled as Co (1) and Co (2). **b** The honeycomb layer viewed along the crystal $c$ axis. The Co²⁺ ions are surrounded by oxygen octahedra. $a$ denotes the zigzag direction (perpendicular to the Co-Co bond), $a^*$ denotes the armchair direction (parallel to the Co-Co bond). Schematic of the thermal transport measurement setup with (**c**) $H // J$ and (**d**) $H \perp J$ (see Methods). Directions of both thermal current $J$ and external magnetic field $H$ are shown with colored arrows.

the $x$ direction. In the monoclinic $\alpha$-RuCl₃ (space group *C2/m*), the honeycomb (*ab*) plane is not a mirror plane, nor is the plane normal to the $a$ axis. Furthermore, the $C_2$ rotational symmetry is broken along the $a$ axis, so a planar $\kappa_{xy}$ is allowed by symmetry for $H // a$, and is indeed observed[10,11]. However, the plane normal to the $b$ direction is a mirror plane and the $C_2$ rotational symmetry is also preserved along the $b$ direction; consistently, measurements report $\kappa_{xy} \simeq 0$ for $H // b$[10].

Here we turn to another Kitaev magnet candidate, the insulating material Na₂Co₂TeO₆[15,16], and present a study of its planar thermal Hall effect. We observe a non-zero planar $\kappa_{xy}$ in Na₂Co₂TeO₆ single crystals. On the basis of a striking similarity between the temperature and field dependence of planar $\kappa_{xy}$ and that of the phonon-dominated $\kappa_{xx}$, we argue that the planar thermal Hall effect in Na₂Co₂TeO₆ is carried predominantly by phonons. We perform a complete study with different in-plane configurations of the heat current $J$ and magnetic field $H$, i.e. $H // J$ and $H \perp J$, and observe a large difference in $\kappa_{xy}$ between these two configurations, which reveals that the direction of the heat current $J$ may play an important role in determining the planar thermal Hall effect. We also observe that the planar $\kappa_{xy}$ shows a strong sample dependence, which imposes a constraint on the mechanism responsible for the phonon thermal Hall effect.

## Results

Na₂Co₂TeO₆ is a honeycomb-layered insulator (Fig. 1a) that develops long-range AF order below $T_N \simeq 27$ K[17] – which resembles the low-temperature formation of AF order in $\alpha$-RuCl₃. It has been theoretically predicted that the Kitaev model can also be realized in materials with $d^7$ ions such as Co²⁺ [18–20] and magnetic excitations in Na₂Co₂TeO₆

indeed resemble calculations based on extended Kitaev-Heisenberg models[21–26]. In our thermal transport study, the magnetic field $H$ and heat current $J$ are both applied in the *ab* plane, either parallel to each other (Fig. 1c) or perpendicular to each other (Fig. 1d). $\kappa_{xx}$ and $\kappa_{xy}$ are measured simultaneously, for four configurations: $H // J // a$ (perpendicular to the Co-Co bond direction), $H // J // a^*$ (parallel to the Co-Co bond direction), $J // a$ & $H // a^*$, and $J // a^*$ & $H // a$. Note that the structure of Na₂Co₂TeO₆ is such that a non-zero $\kappa_{xy}$ is not allowed for either $H // a$ or $H // a^*$ because its crystal structure (space group $P6_322$) has $C_2$ rotational symmetry along both $a$ and $a^*$ directions.

First, we measured $\kappa_{xx}$ and $\kappa_{xy}$ in the two planar-parallel configurations, i.e. $H // J // a$ and $H // J // a^*$ (as shown in Fig. 1c). In Fig. 2a, we show the thermal conductivity $\kappa_{xx}$ of Na₂Co₂TeO₆ as a function of temperature, measured in sample A with configuration $H // J // a$, for $H = 0$, 5, 10 and 15 T. When $H \leq 10$ T, $\kappa_{xx}$ shows little field dependence. Applying 15 T, however, produces a dramatic enhancement of $\kappa_{xx}$ at low $T$, in agreement with prior data[27,28]. A similar behaviour is observed in $\alpha$-RuCl₃[29,30], with a sudden increase of $\kappa_{xx}$ when $H > 7$ T. In both materials, $\kappa_{xx}$ is attributed to phonons that are strongly scattered by spin fluctuations. When a field large enough to suppress AF order is applied in the 2D plane, a spin gap opens in the field-polarized state[31], and so the spin scattering is reduced at low $T$, leading to an increase in $\kappa_{xx}$[27,29,30].

In Fig. 2b, we show the thermal Hall conductivity of Na₂Co₂TeO₆, measured on the same sample (A) in the same configuration ($H // J // a$), plotted as $\kappa_{xy}$ vs $T$ (To obtain $\kappa_{xy}$ data, we use $\kappa_{yy}$ data in Supplementary Fig. 1; see METHODS). Surprisingly, we observe a non-zero $\kappa_{xy}$, which is supposed to be forbidden by the two-fold rotational symmetry along this direction. $\kappa_{xy}(T)$ mirrors the evolution of $\kappa_{xx}(T)$ at

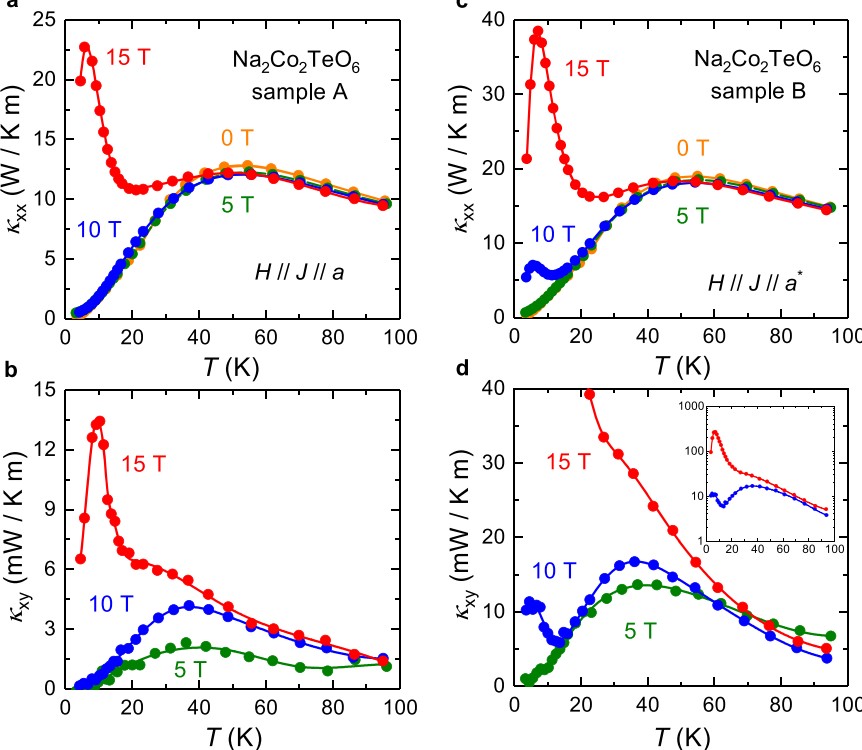

**Fig. 2 | Thermal transport in Na$_2$Co$_2$TeO$_6$ with H // J // a and H // J // a\*.** Thermal conductivity $\kappa_{xx}$ vs temperature $T$ in Na$_2$Co$_2$TeO$_6$ (**a**) sample A measured with $H // J$ // a and (**c**) sample B measured with $H // J // a*$ at $H = 0$ T, 5 T, 10 T, and 15 T, with $J$ the thermal current and $H$ the external magnetic field. Thermal Hall conductivity $\kappa_{xy}$ vs $T$ in Na$_2$Co$_2$TeO$_6$ (**b**) sample A and (**d**) sample B measured at $H = 5$ T, 10 T, and 15 T. In panel (**d**), the inset shows the full range of data. $a$ denotes the zigzag direction (perpendicular to the Co-Co bond), $a*$ denotes the armchair direction (parallel to the Co-Co bond).

different fields. At $H = 5$ T and 10 T, $\kappa_{xy}(T)$ and $\kappa_{xx}(T)$ both show a broad hump around 40 K and decrease monotonically to zero with decreasing temperature, while at $H = 15$ T, both display the same dramatic increase at low $T$, peaking at $T \sim 10$ K. This striking similarity between $\kappa_{xy}(T)$ and $\kappa_{xx}(T)$ is compelling evidence that $\kappa_{xy}$ is carried predominantly by phonons in Na$_2$Co$_2$TeO$_6$. Evidence from other insulators has indeed shown that for phonons $\kappa_{xy}$ and $\kappa_{xx}$ both increase in tandem[32–35]. Our experimental results do not exclude the possibility that exotic neutral excitations such as Majorana fermions or chiral magnons could contribute to the planar $\kappa_{xy}$ signal in Na$_2$Co$_2$TeO$_6$ to some extent. However, our data strongly suggest that phonons are the dominant carriers for the planar thermal Hall effect in this material.

In Fig. 2c and 2d, we report the equivalent study for the planar-parallel configuration $H // J // a*$, performed on a second sample (B). Again, we observe a non-zero $\kappa_{xy}$ signal with $H // J // a*$, which is also supposed to be forbidden by the two-fold rotational symmetry along this direction. This observation shows that the actual mechanism behind the planar thermal Hall effect in this material remains effective even though the pristine lattice has two-fold rotational symmetry in the $a*$ direction. We see again a dramatic increase of both $\kappa_{xy}$ and $\kappa_{xx}$ when a field of 15 T is applied, reinforcing the close correlation between $\kappa_{xy}$ and $\kappa_{xx}$ seen in the first configuration. Interestingly, we find that in the second configuration ($H // a*$) the parallel increase of $\kappa_{xy}$ and $\kappa_{xx}$ at low $T$ even begins at 10 T, further confirming that $\kappa_{xy}$ mimics $\kappa_{xx}$. We infer that the critical field for suppressing the AF order in Na$_2$Co$_2$TeO$_6$ is slightly less than 10 T for $H // a*$ (and more than 10 T for $H // a$), as shown by a previous study of $\kappa_{xx}$[27].

To check the reproducibility of our data, we performed the same measurements on another two samples (C and D) that were cut from the same mother sample. $\kappa_{xx}$ and $\kappa_{xy}$ measured on sample C with $H // J$ // a and on sample D with $H // J // a*$ are plotted in Supplementary Fig. 2.

Similar behavior of $\kappa_{xx}$ and $\kappa_{xy}$ are observed in samples C and D. However, the magnitude of $\kappa_{xy}$ shows a clear sample dependence, especially when comparing samples B and D, which points to an extrinsic origin of the planar thermal Hall effect in Na$_2$Co$_2$TeO$_6$. This sample dependence may also explain the much smaller magnitude of $\kappa_{xy}$ reported in a prior study by Takeda *et al.*[28].

Based on the close similarity we observe – for the two distinct field directions – between the temperature and field dependence of the planar $\kappa_{xy}$ and that of the phonon-dominated $\kappa_{xx}$, we conclude that phonons are responsible for the planar thermal Hall conductivity $\kappa_{xy}$ in Na$_2$Co$_2$TeO$_6$ – where field and current are both in the plane and parallel to each other. This shows it is possible – and makes it likely – that the planar $\kappa_{xy}$ observed in $\alpha$-RuCl$_3$ is also carried by phonons.

In Fig. 3a, we compare the ratio of $\kappa_{xy}$ over $\kappa_{xx}$, plotted as $\kappa_{xy}$ / $\kappa_{xx}$ vs $T$, in all four samples. The ratio of $\kappa_{xy}$ over $\kappa_{xx}$ clearly shows a temperature dependence, which has indeed been observed in several insulators where phonons are responsible for the thermal Hall effect[33,34]. This temperature dependence relates to the dominant scattering mechanism of phonons at different temperature regions. For example, phonon-phonon scattering dominates at high temperatures. As the temperature gradually decreases, other scattering mechanisms start to kick in. Phonons are strongly scattered by impurities and defects around 20 K. As $T$ approaches 0 K, phonons are mainly scattered by the boundaries. This detailed $T$ dependence of the scattering mechanisms causes this $T$ dependence in the ratio of $\kappa_{xy}$ over $\kappa_{xx}$. With a configuration of $H // J // a$, the ratio of sample C is about two times larger than that of sample A, at $T = 20$ K. With a configuration of $H // J // a*$, the ratio of sample B is about five times larger than that of sample D. Although a clear sample dependence is observed, the magnitude of $| \kappa_{xy} / \kappa_{xx} |$ in all cases is typical of the phonon thermal Hall effect found in various insulators (albeit for $H // z$)[33–35], where $0.05\% \lesssim \left| \frac{\kappa_{xy}}{\kappa_{xx}} \right| \lesssim 0.5\%$ at $T = 20$ K and $H = 15$ T. In

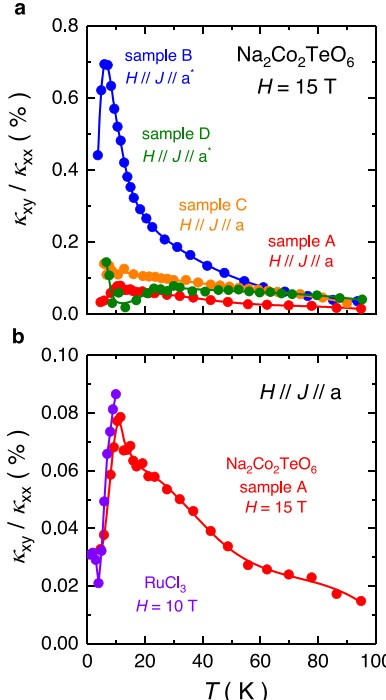

**Fig. 3 | Ratio of $\kappa_{xy}/\kappa_{xx}$ for candidate Kitaev magnets. a** Ratio between the thermal Hall conductivity and thermal conductivity $\kappa_{xy}/\kappa_{xx}$ vs temperature $T$ of the planar thermal Hall response in Na$_2$Co$_2$TeO$_6$ at a magnetic field $H = 15$ T of the four measured samples. Although the ratio clearly shows a sample dependence, the order of magnitude (0.1% at $H = 15$ T and $T = 20$ K) is typical of the phonon thermal Hall effect in various insulators[33–35]. **b** Ratio $\kappa_{xy}/\kappa_{xx}$ vs $T$ of the planar thermal Hall response in candidate Kitaev magnets Na$_2$Co$_2$TeO$_6$ and $\alpha$-RuCl$_3$. The purple curve is obtained from data in Fig. 3.28 of ref. [36]. $a$ denotes the zigzag direction (perpendicular to the Co-Co bond), $a^*$ denotes the armchair direction (parallel to the Co-Co bond).

Fig. 3b, we also see that the planar thermal Hall effect in Na$_2$Co$_2$TeO$_6$ is comparable – in both magnitude and temperature dependence – to that seen in $\alpha$-RuCl$_3$ for $H // J // a$[36]. This striking similarity between Na$_2$Co$_2$TeO$_6$ and $\alpha$-RuCl$_3$ points to a common underlying mechanism.

After measuring both $\kappa_{xx}$ and $\kappa_{xy}$ with the heat current and magnetic field parallel to each other, we conduct the same measurements with $H$ and $J$ both in plane but perpendicular to each other ($H \perp J$). In Fig. 4a, we show the thermal conductivity $\kappa_{xx}$ of Na$_2$Co$_2$TeO$_6$ at $H = 15$ T as a function of temperature with $H // a$, for two current directions: $J // H$ (sample C, $J // a$, red) and $J \perp H$ (sample D, $J // a^*$, blue). In Fig. 4c, we show the same comparison of current directions for $H // a^*$. We see that with the same field direction, $\kappa_{xx}$ for $J \perp H$ is very similar in magnitude and temperature dependence to $\kappa_{xx}$ for $J // H$. In other words, the current direction matters very little for $\kappa_{xx}$. We expect the field direction to matter for $\kappa_{xx}$ because the field affects the magnetism and the spins that scatter phonons, and this effect can in principle be different for $H // a$ and $H // a^*$. On the other hand, if the field direction is kept fixed, changing the current direction from $J // H$ to $J \perp H$ should make little difference to $\kappa_{xx}$. By contrast, $\kappa_{xy}$ decreases dramatically when $J$ changes from being parallel to $H$ to being perpendicular to $H$, as shown in Fig. 4b and d. Unlike $\kappa_{xx}$, the thermal Hall effect is expected to depend crucially on the directions of both $H$ and $J$, relative to the direction $y$ along which $dT_y$ is measured. For example, we expect $dT_y = 0$ when $H // y$, which does appear to be the case in our data on Na$_2$Co$_2$TeO$_6$. This observation clearly shows that the magnitude of the planar $\kappa_{xy}$ strongly depends on the direction of the heat current relative to the magnetic field, i.e. whether $J // H$ or $J \perp H$. A similar behavior is also observed in sample A and sample B (Supplementary Fig. 3).

## Discussion

Three main questions arise. First, what makes phonons chiral in Na$_2$Co$_2$TeO$_6$? The sample dependence we observe suggests an extrinsic origin for the phonon thermal Hall effect, e.g. from scattering of phonons by defects or impurities. In a recent model, it was shown that defects embedded in an insulator with AF order can scatter phonons in a way that produces a thermal Hall effect in a magnetic field[37] – a mechanism that may well explain the dependence of $\kappa_{xy}$ on impurity concentration in the AF insulator Sr$_2$IrO$_4$[38], and perhaps also in cuprates[39,40].

The second question is: how can there be a non-zero $\kappa_{xy}$ signal in Na$_2$Co$_2$TeO$_6$ when $H // J // a$ or $H // J // a^*$, two field directions for which a non-zero $\kappa_{xy}$ is in principle forbidden by the $C_2$ rotational symmetry? Clearly, this planar Hall effect cannot originate from some intrinsic scenario controlled entirely by the underlying crystal symmetry, such as the existing theoretical scenarios for Majorana fermions or topological magnons in $\alpha$-RuCl$_3$. We suggest that the planar $\kappa_{xy}$ may be due to a local symmetry breaking induced by some extrinsic effects. For example, by structural defects like stacking faults or domains, reminiscent of the proposal that structural domains play a role in generating a phonon thermal Hall effect in SrTiO$_3$[32]. Indeed, it has been reported that the Na layers in Na$_2$Co$_2$TeO$_6$ are highly disordered[16], which could possibly break the local crystal symmetry.

The third question is: how to understand the large difference in the magnitude of $\kappa_{xy}$ between $J // H$ and $J \perp H$? Our results indicate that when putting both $H$ and $J$ in the plane, the planar $\kappa_{xy}$ can be dramatically reduced when current and field are perpendicular to each other. In previous theoretical explanations[10,12,13] for the planar thermal Hall effect observed in $\alpha$-RuCl$_3$, whether a non-zero planar $\kappa_{xy}$ can arise only depends on the underlying crystal symmetry and the direction of magnetic field, regardless of the direction of heat current. Our findings reveal that the direction of heat current also plays an important role in producing the planar thermal Hall effect. This calls for a re-evaluation of the mechanism responsible for the planar $\kappa_{xy}$ observed in $\alpha$-RuCl$_3$.

Note that in addition to Na$_2$Co$_2$TeO$_6$, we have also observed a phononic planar thermal Hall signal (comparable in magnitude to the conventional thermal Hall signal) in both cuprates[41] and in the frustrated antiferromagnetic insulator Y-kapellasite[42], thereby further validating the existence of a planar thermal Hall signal coming from phonons.

## Methods
### Samples
Single crystals of Na$_2$Co$_2$TeO$_6$ were grown by a self-flux method starting from Na$_2$CO$_3$, Co$_3$O$_4$ and TeO$_2$ in a molar ratio of 15.4: 5.2: 21.4. These oxides were ground thoroughly and put into an alumina crucible, which was then heated to 1323 K in 4 hours and maintained for 48 hours before being cooled down to 873 K in 6.5 K/hour. The furnace was turned off at 873 K to cool down to room temperature. Thin single crystals of hexagonal shape were harvested from the solidified flux. The edge of the hexagon is along the $a$ axis of the crystal structure.

Four single crystal samples of Na$_2$Co$_2$TeO$_6$ were used in the heat transport measurements. Sample A has dimensions $L = 0.95$ mm (length between contacts, along $x$), $w = 1.62$ mm (width, along $y$) and $t = 0.1$ mm (thickness, along $z$), with the $x$ direction (Fig. 1c) along the $a$ axis of the crystal structure (Fig. 1b). Sample B has dimensions $L = 1.55$ mm (along $x$), $w = 2.83$ mm (along $y$) and $t = 0.07$ mm (along $z$), with the $x$ direction along the $a^*$ axis of the crystal structure (Fig. 1b). Sample C has dimensions $L = 0.84$ mm (length between contacts, along $x$), $w = 1.00$ mm (width, along $y$) and $t = 0.04$ mm (thickness, along $z$), with the $x$ direction (Fig. 1c) along the $a$ axis of the crystal structure (Fig. 1b). Sample D has dimensions $L = 1.43$ mm (along $x$), $w = 0.73$ mm (along $y$) and $t = 0.05$ mm (along $z$), with the $x$ direction along the $a^*$

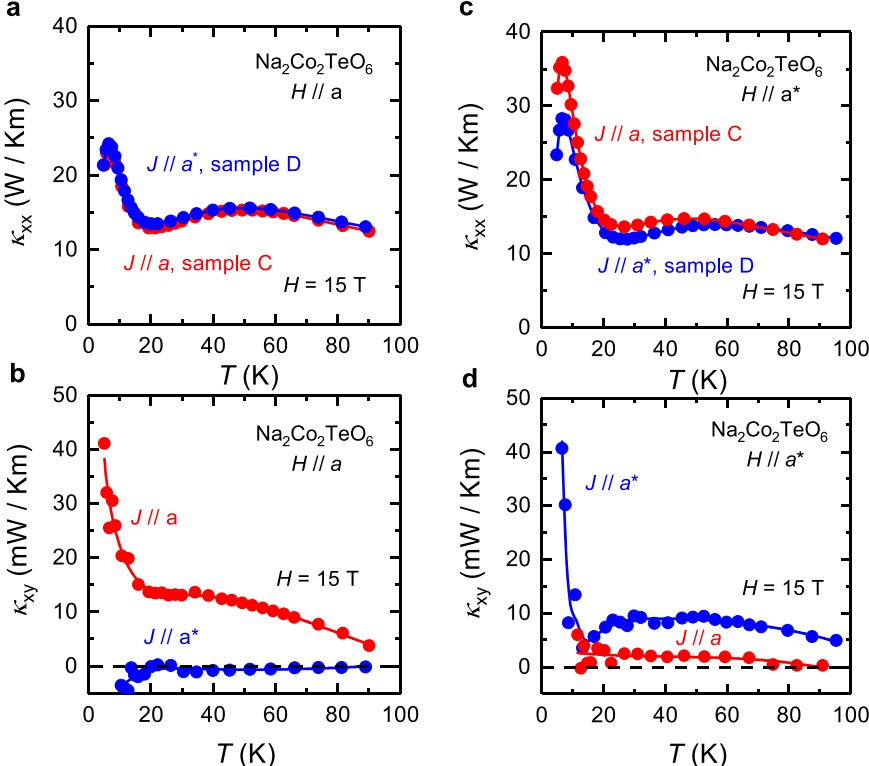

**Fig. 4 | Thermal transport data in Na₂Co₂TeO₆ for *H* // *J* and *H* ⊥ *J*. a** Thermal conductivity $\kappa_{xx}$ and (**b**) thermal Hall conductivity $\kappa_{xy}$ vs temperature *T* in Na₂Co₂TeO₆, measured at a magnetic field *H* = 15 T for *H* // *a*: on sample C with *J* // *a* (red) and sample D with *J* // *a** (blue). **c**, **d** Corresponding data for *H* // *a**. In both field directions, $\kappa_{xy}$ measured with *H* ⊥ *J* is much smaller than that measured with *H* // *J*. *a* denotes the zigzag direction (perpendicular to the Co-Co bond), *a** denotes the armchair direction (parallel to the Co-Co bond).

axis of the crystal structure (Fig. 1b). Sample A and B are two separate as-grown samples, while sample C and sample D are cut from one as-grown mother sample that is from the same batch of A and B.

Contacts were made by attaching 50 *μm* diameter silver wires to the sample using silver paint. The heater was connected to the sample by 100 *μm* diameter silver wire by silver paint.

**Thermal transport measurements**

The thermal conductivity $\kappa_{xx}$ and thermal Hall conductivity $\kappa_{xy}$ were measured by applying a heat current *J* along the length of the sample (*J* // *x*; Fig. 1c and d) and a magnetic field *H* parallel to *J* (*H* // *x*; Fig. 1c) or perpendicular to *J* (*H* // *y*; Fig. 1d), in the so-called "planar" configuration. The current produces a longitudinal temperature difference $\Delta T_x$ along *x* (between the two contacts separated by the distance *L*). The thermal conductivity is defined as $\kappa_{xx} = (J/\Delta T_x)(L/wt)$. The field produces a transverse temperature difference $\Delta T_y$ along *y* (between the two sides of the sample, separated by the sample width *w*). The thermal Hall conductivity is defined as $\kappa_{xy} = -\kappa_{yy}(\Delta T_y/\Delta T_x)(L/w)$.

For sample A, the current and field directions are *J* // *H* // *a* or *J* // *a* & *H* // *a**, where *a* is perpendicular to the Co-Co bond direction in the lattice and *a** is parallel to the Co-Co bond direction. For sample B, *J* // *H* // *a** or *J* // *a** & *H* // *a*. In a honeycomb lattice, $\kappa_{xx} \neq \kappa_{yy}$. For sample A and B, we obtain $\kappa_{yy}$ by multiplying the $\kappa_{xx}$ measured on the same sample by the anisotropy factor $\kappa_{yy}/\kappa_{xx}$ reported in ref. 27. (see Supplementary Fig. 1). $\kappa_{yy}$ and $\kappa_{xx}$ reported in ref. 27 are measured on two samples that are cut from the same mother sample, which reflects the intrinsic anisotropy of the longitudinal thermal conductivity when the heat current is applied along *a* or *a** direction. This anisotropy is also consistent with what we get from sample C and sample D, which are cut from the same mother samples.

The experimental technique used here is as follows. The heat current is generated by a resistive heater connected to one end of the sample (Fig. 1c and d). The other end of the sample is glued to a copper block with silver paint that acts as a heat sink. The longitudinal and transverse temperature differences $\Delta T_x$ and $\Delta T_y$ are measured using type-E thermocouples. All the measurements are conducted with a steady-state method in a variable temperature insert (VTI) system up to *H* = 15 T. The data were taken by changing temperature in discrete steps at a fixed magnetic field. After the temperature is stabilized at each temperature, the background value of the thermocouple is eliminated by subtracting the heater-off value from the heater-on value. When measuring $\kappa_{xy}$, the contamination from $\kappa_{xx}$ due to a slight misalignment of contacts for $\Delta T_y$ is removed by doing field anti-symmetrization to the transverse temperature difference. That is to say, we measure $\Delta T_y$ with both positive and negative magnetic fields exactly in the same conditions, then the transverse temperature difference used to obtain $\kappa_{xy}$ is defined as $\Delta T_y(H) = [\Delta T_y(T, +H) - \Delta T_y(T, -H)]/2$.

**Thermal conductivity and thermal hall conductivity measurements in four Na₂Co₂TeO₆ samples**

$\kappa_{xx}$ and $\kappa_{xy}$ measured in sample C with *H* // *J* // *a* and sample D with *H* // *J* // *a** are plotted in Supplementary Fig. 2.

$\kappa_{xx}$ and $\kappa_{xy}$ measured in sample A with *H* // *J* // *a* or *J* // *a* & *H* // *a** and sample B with *H* // *J* // *a** or *J* // *a** & *H* // *a* are plotted in Supplementary Fig. 3.

## Data availability

All the data that support the findings of this study are available from the corresponding authors upon request.

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

## Acknowledgements

L.T. acknowledges support from the Canadian Institute for Advanced Research (CIFAR) as a Fellow and funding from the Natural Sciences and Engineering Research Council of Canada (NSERC; PIN: 123817), the Fonds de recherche du Québec - Nature et Technologies (FRQNT), the Canada Foundation for Innovation (CFI), and a Canada Research Chair. This research was undertaken thanks in part to funding from the Canada First Research Excellence Fund. The work at Peking University was supported by the National Basic Research Program of China (Grant No. 2021YFA1401901) and the NSF of China (Grant No. 12061131004).

## Author contributions

W.Y. and Y.L. grew the Na$_2$Co$_2$TeO$_6$ single crystals. L.C. and A.V. prepared the samples. L.C., É.L., A.V., A.A and Q.B. performed the thermal transport measurements. L.C. and L.T. wrote the manuscript, in consultation with all the authors. L.T. supervised the project.

## Competing interests

The authors declare no competing interests.
