## [Peer Review File · Nature Communications]

REVIEWER COMMENTS

Reviewer #1 (Remarks to the Author):

The authors investigated thermal transport in the Mott insulator $\text{Na}_2\text{Co}_2\text{TeO}_6$, which is known as a Kitaev magnet candidate. In particular, they focused on a planar thermal Hall effect. This effect has attracted considerable attention since the observation of the half-quantized κ_{xy} plateau in the Kitaev candidate $\alpha\text{-RuCl}_3$. The half quantization provides compelling evidence of Majorana fermions as heat carriers. On the other hand, it has been argued that topological magnons can also generate a planar thermal Hall effect without half quantization. Thus far, the origin of such a thermal Hall effect remains controversial. Phonons also potentially give rise to a thermal Hall effect. In their study, the authors explored the possibility of phonons causing the planar thermal Hall effect in $\text{Na}_2\text{Co}_2\text{TeO}_6$. They conducted a comprehensive examination of the longitudinal and transverse thermal conductivities by preparing several experimental configurations for measuring thermal transport. They also measured the thermal transport in four samples to check the reproducibility of their data. In $\text{Na}_2\text{Co}_2\text{TeO}_6$, a nonzero κ_{xy} by an inplane magnetic field is prohibited due to its crystal symmetry. Nevertheless, they observed a planar thermal Hall effect. They also found a similarity between κ_{xx} and κ_{xy} , which they consider compelling evidence for the predominant contribution of phonons to the planar thermal Hall effect. Moreover, the authors demonstrated that the magnitude of the thermal Hall conductivity depends on the configuration of the heat current and magnetic field; κ_{xy} is strongly suppressed when $\mathbf{H} \perp \mathbf{J}$. In summary, this is a well-written paper, which shows novel and interesting results. In particular, the dependence of κ_{xy} on the relative configuration of \mathbf{H} and \mathbf{J} is intriguing and provides an important clue to understanding the origin of the thermal Hall effect. However, I recommend the authors provide more detailed information and/or explanations to clarify why the thermal transport of $\text{Na}_2\text{Co}_2\text{TeO}_6$ originates from phonons. Therefore, the authors should address the following issues properly before I recommend publication.

- On page 5, the authors claimed that "this striking similarity between $\kappa_{xy}(T)$ and $\kappa_{xx}(T)$ is compelling evidence that κ_{xy} is carried predominantly by phonons in $\text{Na}_2\text{Co}_2\text{TeO}_6$. By contrast, if κ_{xy} were caused by Majorana fermions or magnons (or any other spin-based excitation), there would be no reason for it to mimic the phonon-dominated $\kappa_{xx}(T)$." From these sentences, the authors do not perfectly deny that the main origin of κ_{xy} is Majorana fermions or magnons. However, the authors concluded that the experimental results are compelling evidence that phonons mainly govern the thermal Hall effect. I cannot understand the logic of concluding a phonon Hall effect. If the authors claim a phonon Hall effect, they should provide experimental evidence that completely refutes the origin of Majorana fermions and magnons.

- Other than the similarity between κ_{xx} and κ_{xy} , the authors should mention in their paper the microscopic origin of the thermal Hall effect that causes phonon to produce a thermal Hall effect of about $\kappa \sim 10 \text{ mW/Km}$. Also, the presence of phonons and impurities is ubiquitous in ordinary

insulators; if we consider impurity scattering to phonons to be the origin, does the $\kappa_{xy} \sim 10 \text{ mW/Km}$ universally occur? Or does it predict the presence of some special phonons in $\text{Na}_2\text{Co}_2\text{TeO}_6$?

- From Fig. 3, we find a large sample and temperature dependence of $|\kappa_{xy} / \kappa_{xx}|$. This appears to contradict the similarity between κ_{xx} and κ_{xy} , which should also be mentioned. If a nonzero planar thermal Hall conductivity originates from extrinsic effects, is it correct to conclude that phonons are the origin of the thermal Hall effect based solely on the similarity?

- On page 8, the authors claimed, "Clearly, this planar Hall effect cannot originate from some intrinsic scenario controlled entirely by the underlying crystal symmetry, such as the scenario of Majorana fermions or topological magnons." Related to this statement, why does prohibiting κ_{xy} suggest only the extrinsic effects of phonons and deny extrinsic effects on Majorana fermions or topological magnons, such as impurity scatterings for these quasiparticles?

- The dependence of κ_{xy} on the relative configuration of the magnetic field and heat current is very interesting. However, there appears to be a lack of discussion of its relationship to the authors' claim that it is of phonon origin. It has been shown that κ_{xx} shows no such relative configuration dependence, as shown in Figs. 4a and 4c. This appears to contradict the similarity between κ_{xx} and κ_{xy} , which is the basis for the phonon origin, but the authors should discuss the discrepancy.

Reviewer #2 (Remarks to the Author):

The manuscript presents a study on the planar thermal Hall effect in a Kitaev candidate material, $\text{Na}_2\text{Co}_2\text{TeO}_6$. The authors have carefully conducted the planar thermal Hall effect experiments on four specimens using four distinct orientational dependent configurations. Compared their finding with previous results in another Kitaev candidate, $\alpha\text{-RuCl}_3$, the authors argue that the planar thermal Hall anomaly of $\text{Na}_2\text{Co}_2\text{TeO}_6$ can be attributed to phonon contribution. Their experimental methodology and investigations into the planar Hall effect are thorough and convincing.

However, despite the experimental strengths and the clear presentation of the study, the manuscript does not seem to achieve a level to deliver groundbreaking or significance to the field, compared with the previous studies [PRB 104, 14426(2021); PRB 106, L081116 (2022); PRS 4, L042035 (2022); PRB 107, 184423 (2023)]. Thus, the current findings do not appear to provide a significant leap beyond these earlier works. This lack of groundbreaking contributions or substantial advancements makes the manuscript less suitable for publication in Nature Communications.

Reviewer #3 (Remarks to the Author):

Recently, thermal Hall effect of quantum magnets has received extensive research interests. An exciting experimental finding is the half-quantized thermal Hall conductivity κ_{xy} in the Kitaev antiferromagnet α -RuCl₃ for an in-plane field in excess of 7 T, which was interpreted as evidence of itinerant Majorana fermions. The half-quantized κ_{xy} plateau appears even for a planar thermal Hall configuration, i.e., when the magnetic field is applied within the 2D plane and parallel to the heat current J ($J // H // a$, where a is the crystal direction perpendicular to the Ru-Ru bond, the so-called zigzag direction). However, this phenomenon has not been fully confirmed and the magnons and phonons were also proposed to contribute to the planar thermal Hall effect of this famous material.

In this work, Chen et al. studied the planar thermal Hall effect of another Kitaev material Na₂Co₂TeO₆. The main experimental findings include: (i) there are non-zero κ_{xy} for either $J // H // a$ or $J // H // a^*$, which are proposed to be forbidden by the two-fold rotational symmetry along these directions; (ii) there is close similarity between the temperature and field dependence of the planar κ_{xy} and those of phonon-dominated κ_{xx} ; (iii) the planar κ_{xy} strongly depends on the direction of the heat current relative to the magnetic field, i.e., $J // H$ or J perpendicular to H . The main conclusion is that phonons are responsible for the planar thermal Hall conductivity.

In general, this work displays some peculiar experimental results, like the non-zero planar thermal Hall conductivity forbidden by the two-fold rotational symmetry. The conclusion, if correct, would be very important for understanding the present topic on the origin of thermal Hall effect in the Kitaev system. The paper was well written and is suitable for Nature Communications. However, I have one comment to be considered by the authors.

The authors showed the temperature dependence of κ_{xy} and κ_{xx} in a few different magnetic fields and made the comparison. The key experimental result the authors concluded is the close similarity in the temperature and field dependence between κ_{xy} and κ_{xx} . Here, the question is: what is the detailed magnetic field dependence of κ_{xy} and κ_{xx} ? Probably, no material can display similar magnetic field dependence of κ_{xy} between κ_{xx} because of their different origins. Nevertheless, without the comparison of $\kappa_{xy}(H)$ and $\kappa_{xx}(H)$ data, it is questionable to claim the similarity of magnetic field dependence between them.

POINT-BY-POINT RESPONSE TO REVIEWER COMMENTS

Reviewer #1

The authors investigated thermal transport in the Mott insulator $\text{Na}_2\text{Co}_2\text{TeO}_6$, which is known as a Kitaev magnet candidate. In particular, they focused on a planar thermal Hall effect. This effect has attracted considerable attention since the observation of the half-quantized K_{xy} plateau in the Kitaev candidate $\alpha\text{-RuCl}_3$. The half quantization provides compelling evidence of Majorana fermions as heat carriers. On the other hand, it has been argued that topological magnons can also generate a planar thermal Hall effect without half quantization. Thus far, the origin of such a thermal Hall effect remains controversial. Phonons also potentially give rise to a thermal Hall effect. In their study, the authors explored the possibility of phonons causing the planar thermal Hall effect in $\text{Na}_2\text{Co}_2\text{TeO}_6$. They conducted a comprehensive examination of the longitudinal and transverse thermal conductivities by preparing several experimental configurations for measuring thermal transport. They also measured the thermal transport in four samples to check the reproducibility of their data. In $\text{Na}_2\text{Co}_2\text{TeO}_6$, a nonzero K_{xx} by an in-plane magnetic field is prohibited due to its crystal symmetry. Nevertheless, they observed a planar thermal Hall effect. They also found a similarity between K_{xx} and K_{xy} , which they consider compelling evidence for the predominant contribution of phonons to the planar thermal Hall effect. Moreover, the authors demonstrated that the magnitude of the thermal Hall conductivity depends on the configuration of the heat current and magnetic field; K_{xy} is strongly suppressed when $H \perp J$.

In summary, this is a well-written paper, which shows novel and interesting results. In particular, the dependence of K_{xy} on the relative configuration of H and J is intriguing and provides an important clue to understanding the origin of the thermal Hall effect.

We thank the Reviewer for recognizing the novelty and interest of our work.

However, I recommend the authors provide more detailed information and/or explanations to clarify why the thermal transport of $\text{Na}_2\text{Co}_2\text{TeO}_6$ originates from phonons. Therefore, the authors should address the following issues properly before I recommend publication.

- On page 5, the authors claimed that "this striking similarity between $K_{xy}(T)$ and $K_{xx}(T)$ is compelling evidence that K_{xx} is carried predominantly by phonons in $\text{Na}_2\text{Co}_2\text{TeO}_6$. By contrast, if K_{xy} were caused by Majorana fermions or magnons (or any other spin-based excitation), there would be no reason for it to mimic the phonon-dominated $K_{xx}(T)$." From these sentences, the authors do not perfectly deny that the main origin of k_{xy} is Majorana fermions or magnons. However, the authors concluded that the experimental results are compelling evidence that phonons mainly govern the thermal Hall effect. I cannot understand the logic of concluding a phonon Hall effect. If the authors claim a phonon Hall effect, they should provide experimental evidence that completely refutes the origin of Majorana fermions and magnons.

We thank the Reviewer for requesting that we further clarify our argument in favor of phonons as the main carriers responsible for the thermal Hall effect in $\text{Na}_2\text{Co}_2\text{TeO}_6$ (NCTO).

We cannot prove that Majorana fermions or topological magnons make no contribution to K_{xy} in $\text{Na}_2\text{Co}_2\text{TeO}_6$. We can only provide arguments for why phonons do make a major contribution. Our argument is based on a detailed comparison of K_{xy} and K_{xx} .

First, we recognize that K_{xx} is dominated by phonons, as was carefully and convincingly demonstrated by Hess and coworkers for both $\alpha\text{-RuCl}_3$ [29] and NCTO [27]. At low temperature, inside the AF phase of NCTO ($T < T_N \sim 27$ K), K_{xx} increases rapidly as soon as a magnetic field large enough to suppress the AF order is applied. The critical field for suppressing AF order in NCTO is slightly more than 10 T for $B // a$ and slightly less than 10 T for $B // a^*$ [27]. Further increasing the field from there simply increases the gap in the high-field paramagnetic phase and this gaps out the low-energy spin excitations that scatter phonons. Hence an increase in B causes an increase in K_{xx} . See Fig. A.

Fig. A. Thermal conductivity K_{xx} of $\text{Na}_2\text{Co}_2\text{TeO}_6$ in various magnetic fields applied along the in-plane high-symmetry directions a^* (a) and a (d). Panels (b) and (e) are a low-temperature zoom of the data in panels (a) and (d), respectively. This figure is reproduced from ref. 27 [Hong *et al.*, Phys. Rev. B **104**, 144426 (2021)].

Secondly, we compare our K_{xy} data to their K_{xx} data, in Fig. B. We observe a striking similarity. At 5 T, both K_{xy} and K_{xx} decrease monotonically to zero with decreasing temperature, whereas at 15 T, both K_{xy} and K_{xx} show a huge peak at low temperature. This is seen for both field directions ($B // a$ and $B // a^*$). At 10 T, there is a clear difference between $B // a$ and $B // a^*$: for $B // a$, both K_{xy} and K_{xx} show a similar monotonic decrease, as for the case of 5 T; for $B // a^*$, however, where the critical field is lower, 10 T is enough to cause the start of a rise at low T in K_{xx} , and, remarkably, this feature is also seen in K_{xy} . The fact that $K_{xy}(T,H)$ mimics the behavior of the phonon-dominated $K_{xx}(T,H)$ is compelling evidence that K_{xy} is also dominated by phonons.

In the revised manuscript, we include a more detailed discussion on the temperature and field dependence of both $K_{xy}(T,H)$ and $K_{xx}(T,H)$ to strengthen our claim of a phonon scenario. We also add the following sentence to the manuscript: “Our experimental results do not exclude the possibility that exotic neutral excitations such as Majorana fermions or chiral magnons could contribute to the planar K_{xy} signal in $\text{Na}_2\text{Co}_2\text{TeO}_6$ to some extent. However, our data strongly suggest that phonons are the dominant carriers for the planar thermal Hall effect in this material.”

Fig. B. Comparison of our thermal Hall conductivity data on the right (panels c and f) to published K_{xx} data on the left (panels a and d; from Fig. A), for the same field directions ($B // a^*$ in the top panels and $B // a$ in the bottom panels).

- Other than the similarity between K_{xx} and K_{xy} , the authors should mention in their paper the microscopic origin of the thermal Hall effect that causes phonon to produce a thermal Hall effect of about $K_{xy} \sim 10$ mW/Km. Also, the presence of phonons and impurities is ubiquitous in ordinary insulators; if we consider impurity scattering to phonons to be the origin, does the $K_{xy} \sim 10$ mW/Km universally occur? Or does it predict the presence of some special phonons in $\text{Na}_2\text{Co}_2\text{TeO}_6$?

The fundamental mechanism responsible for the phonon thermal Hall effect in any material is still an open question, even in the conventional configuration (with an out-of-plane magnetic field perpendicular to the heat current), despite several recent experimental [32,34,35,41,42] and theoretical studies [37,A,B,C]. So far, regarding the mechanism behind the phonon thermal Hall effect, there are two classes of scenarios: scenarios based on the coupling of phonons to their pristine environment and scenarios based on the skew scattering of phonons by impurities or defects. Our results provide a new facet of the phonon thermal Hall effect, which is to say, phonons can also generate a “planar” thermal Hall effect. This new finding will shed light on the microscopic mechanism behind the phonon thermal Hall effect in general.

We agree that the presence of phonons and impurities is ubiquitous in ordinary insulators. However, not every insulating material will have a non-zero phonon thermal Hall effect. Take two closely related pyrochlore materials as an example: $\text{Tb}_2\text{Ti}_2\text{O}_7$ has a sizable K_{xy} signal, whereas the K_{xy} signal in $\text{Y}_2\text{Ti}_2\text{O}_7$ is zero [D]. So, whether there is a non-zero K_{xy} signal in an insulator really depends on the detailed mechanism that causes this phonon thermal Hall effect.

For the particular mechanism of phonons scattered by impurities, the magnitude of K_{xy} depends on many parameters, for example, the impurity level and the surrounding environment that the impurities are embedded within [37]. The study of the conventional thermal Hall conductivity in Sr_2IrO_4 [38] shows that the magnitude of K_{xy} strongly depends on the concentration of Rh impurities. Indeed, the thermal Hall angle $|K_{xy} / K_{xx}|$ in $\text{Sr}_2\text{Ir}_{1-x}\text{Rh}_x\text{O}_4$ increases 70 times when going from $x = 0$ to $x = 0.05$, and subsequently decreases to a very small value at $x = 0.15$, when the AF-ordered phase ends. So $K_{xy} \sim 10$ mW/Km is not a universal value and it depends on many parameters regarding the material itself.

- From Fig. 3, we find a large sample and temperature dependence of $|K_{xy} / K_{xx}|$. This appears to contradict the similarity between K_{xx} and K_{xy} which should also be mentioned. If a nonzero planar thermal Hall conductivity originates from extrinsic effects, is it correct to conclude that phonons are the origin of the thermal Hall effect based solely on the similarity?

For the similarity between K_{xy} and K_{xx} , what we refer to is the similarity between the temperature and field dependence of $K_{xy}(H, T)$ and $K_{xx}(H, T)$, as illustrated in Fig. B. That is to say, when K_{xx} shows a peak at low temperature and at high field, K_{xy} also shows a similar peak within the same field and temperature region. This similarity between K_{xy} and

K_{xx} does not contradict the fact that the ratio between these two values still shows a temperature dependence.

In fact, it has been shown that in many different insulators, where phonons are responsible for the thermal Hall effect, that the thermal Hall angle $|K_{xy} / K_{xx}|$ indeed has a strong temperature dependence. This temperature dependence relates to the dominant scattering mechanism of phonons at different temperature regions. In the cuprate Mott insulators, for example, the ratio $|K_{xy} / K_{xx}|$ is temperature dependent and peaks at around $T \sim 20$ K (see Fig. C). At high temperature, phonon-phonon scattering dominates. As the temperature decreases, other scattering mechanisms start to kick in. Phonons are strongly scattered by impurities and defects around 20 K. As $T \rightarrow 0$ K, phonons are mainly scattered by the boundaries. This detailed T dependence of the scattering mechanisms causes this T dependence in the ratio $|K_{xy} / K_{xx}|$.

Fig. C. Ratio of K_{xy} over K_{xx} in three cuprate Mott insulators (expressed in %), measured in a field of 15 T applied parallel to the c axis: $Sr_2CuO_2Cl_2$ (green), La_2CuO_4 (blue), and Nd_2CuO_4 (red). Figure taken from ref. 33 [Boulanger *et al.*, Nat. Commun. **11**, 5325 (2020)].

Even though the ratio $|K_{xy} / K_{xx}|$ shows a temperature dependence, the magnitude is comparable for most of the materials where phonons dominate the thermal Hall effect, falling within the range $0.1\% < |K_{xy} / K_{xx}| < 1\%$ for $B = 15$ T at $T = 20$ K (see ref. [34]).

Fig. D shows a comparison of K_{xy}/B and K_{xx} in different insulators. Although the longitudinal thermal conductivity K_{xx} varies by 4 orders of magnitude, the ratio $K_{xy} / (K_{xx} B)$ remains within the range $10^{-4} - 10^{-3} \text{ T}^{-1}$, which is a typical value for the phonon thermal Hall effect.

Now looking back at the ratio observed in $\text{Na}_2\text{Co}_2\text{TeO}_6$, although the value shows some sample dependence, it also stays within the range $0.1\% < |K_{xy} / K_{xx}| < 1\%$ (see Fig. 3b), which is consistent with the values found in various insulators where phonons dominate the thermal Hall effect (as shown in Fig. D and ref. 34). This is further support for our interpretation that the planar thermal Hall effect in $\text{Na}_2\text{Co}_2\text{TeO}_6$ is likely to come from phonons.

In the revised manuscript, we give further explanations for the temperature dependence of the ratio $|K_{xy} / K_{xx}|$ found in $\text{Na}_2\text{Co}_2\text{TeO}_6$.

Fig. D. Comparison of K_{xy}/B and K_{xx} in different insulators. Taken from ref. 35 [Li *et al.*, Nat. Commun. **14**, 1027 (2023)].

- On page 8, the authors claimed, "Clearly, this planar Hall effect cannot originate from some intrinsic scenario controlled entirely by the underlying crystal symmetry, such as the scenario of Majorana fermions or topological magnons." Related to this statement, why does prohibiting K_{xy} suggest only the extrinsic effects of phonons and deny extrinsic effects on Majorana fermions or topological magnons, such as impurity scatterings for these quasiparticles?

All the existing theories on the planar thermal Hall effect due to Majorana fermions [8] and topological magnons [6,7] are based on intrinsic mechanisms that satisfy the symmetry requirements of the underlying crystal structure. However, we agree with the Reviewer that our results don't exclude other extrinsic mechanisms based on the scattering of Majorana fermions and topological magnons. We hope that our experimental study will motivate the theorists to further develop the theories regarding the planar thermal Hall effect. But again, the case for a K_{xy} due mostly to phonons is compelling in view of its striking similarity with the phonon-dominated $K_{xx}(T,H)$.

In the revised manuscript, we modified this sentence to be "Clearly, this planar Hall effect cannot originate from some intrinsic scenario controlled entirely by the underlying crystal symmetry, such as the existing theories of Majorana fermions or topological magnons. However, it does not rule out the possibility of extrinsic mechanisms based on Majorana fermions or topological magnons. Although, as we have argued, phonons appear to dominate K_{xy} , it is possible that these other excitations could contribute to some extent to the planar thermal Hall effect in $\text{Na}_2\text{Co}_2\text{TeO}_6$."

- The dependence of K_{xy} on the relative configuration of the magnetic field and heat current is very interesting. However, there appears to be a lack of discussion of its relationship to the authors' claim that it is of phonon origin. It has been shown that K_{xx} shows no such relative configuration dependence, as shown in Figs.4a and 4c. This appears to contradict the similarity between K_{xx} and K_{xy} , which is the basis for the phonon origin, but the authors should discuss the discrepancy.

As seen in Figs. 4a and 4c, K_{xx} does not care much about the current direction – the in-plane anisotropy of the thermal conductivity is small. In principle, the field direction matters for K_{xx} because the field affects the magnetism, and the spins scatter phonons, and this effect can be different for $B // a$ and $B // a^*$. So for a fixed field direction, the fact that we change the current direction from $J // B$ to $J \perp B$ makes little difference, as seen in Figs. 4a and 4c. This is not surprising.

By contrast, the Hall effect (whether electrical or thermal) is expected to depend crucially on the directions of both J and B , relative to the direction y along which the voltage (V_y) or T difference (dT_y) is measured. Specifically, we expect $V_y = 0$ or $dT_y = 0$ when $B // y$. And this does appear to be the case in our data on NCTO, where we observe a very small K_{xy} signal when $J \perp B$ (see Figs. 4b and 4d). This finding is not surprising. Now, we also expect $V_y = 0$ or $dT_y = 0$ when $B // J$. And this is not the case in our thermal Hall data on NCTO, where we observe a large K_{xy} signal when $J // B$ (see Figs. 4b and 4d).

Our finding of a strong dependence of K_{xy} on current direction, in the absence of such a dependence for K_{xx} , argues neither in favor nor against a phonon scenario. It is simply a separate observation: not only do we find a large K_{xy} for $J // B$, we also find a negligible K_{xy} for $J \perp B$. Both findings shed new light on the phonon thermal Hall effect.

Our observation that the planar K_{xy} signal is strongly suppressed for a configuration of both field and current in plane but perpendicular to each other has also been found in the paramagnetic regime of the frustrated antiferromagnetic insulator Y-kapellasite [40], where phonons are unambiguously identified to be the source of the planar thermal Hall effect.

In the revised manuscript, we have enlarged the discussion on the difference between the two experimental configurations.

Reviewer #2

The manuscript presents a study on the planar thermal Hall effect in a Kitaev candidate material, $\text{Na}_2\text{Co}_2\text{TeO}_6$. The authors have carefully conducted the planar thermal Hall effect experiments on four specimens using four distinct orientational dependent configurations. Compared their finding with previous results in another Kitaev candidate, $\alpha\text{-RuCl}_3$, the authors argue that the planar thermal Hall anomaly of $\text{Na}_2\text{Co}_2\text{TeO}_6$ can be attributed to phonon contribution. Their experimental methodology and investigations into the planar Hall effect are thorough and convincing.

We thank the Reviewer for recognizing the thoroughness and convincingness of our work.

However, despite the experimental strengths and the clear presentation of the study, the manuscript does not seem to achieve a level to deliver groundbreaking or significance to the field, compared with the previous studies [PRB 104, 144426 (2021); PRB 106, L081116 (2022); PRS 4, L042035 (2022); PRB 107, 184423 (2023)]. Thus, the current findings do not appear to provide a significant leap beyond these earlier works. This lack of groundbreaking contributions or substantial advancements makes the manuscript less suitable for publication in Nature Communications.

We thank the Reviewer for recognizing the experimental strengths and clear presentation of our work. However, we disagree with the reviewer's comment that this work does not deliver groundbreaking or significant findings compared with previous studies. Here, please allow us to further explain the difference between our findings and those of previous thermal transport studies in $\text{Na}_2\text{Co}_2\text{TeO}_6$.

First, two of the previous studies mentioned by the Reviewer – PRB 104, 144426 (2021) and PRB 107, 184423 (2023) – only involve the thermal conductivity K_{xx} of $\text{Na}_2\text{Co}_2\text{TeO}_6$, and do not present any thermal Hall (K_{xy}) data.

Secondly, the previous study PRB 106, L081116 (2022) only involves the *conventional* thermal Hall effect, i.e. with a magnetic field applied along the c axis ($H // c$). What has been carried out in our study is the *planar* thermal Hall effect, i.e. with a magnetic field applied parallel to the layers ($H \perp c$), either parallel or perpendicular to the heat current. Compared to the conventional thermal Hall effect, the planar thermal Hall effect has been studied in much fewer materials (so far, only in the Kitaev candidate materials α -RuCl₃ and Na₂Co₂TeO₆) and lacks thorough understanding. The novelty of our work is that we report data on the planar thermal Hall effect in NCTO.

The only previous work on the planar thermal Hall effect in Na₂Co₂TeO₆ is that of Takeda *et al.* [Phys. Rev. Research 4, L042035 (2022)]. Although both our study and theirs report a non-zero planar K_{xy} in the configuration of $H // J$, the completeness of the studies, the detailed features of the data, and the theoretical explanations are very different in the two studies.

Here are the major differences between our study and that of Takeda *et al.*:

1. The magnitude of the planar K_{xy} in the data of Takeda *et al.* is ~ 10 times smaller than what we observed, possibly due to the sample dependence that has also been seen in our study.
2. In the contact configuration used by Takeda *et al.*, the two contacts that were used to measure the transverse temperature difference dT_y are very close to the heat sink. This can potentially (thermally) short the transverse temperature gradient, thereby explaining the very small K_{xy} signal observed in their measurement.
3. Takeda *et al.* saw a sign change in their planar K_{xy} signal as a function of magnetic field, while our data do not show any sign change, in any of our four samples.
4. Takeda *et al.* did their measurements for only the configuration of $H // J$, whereas we performed a more complete study, including both $H // J$ and $H \perp J$ (in-plane). The extra configuration provides important information about the dependence of the planar K_{xy} on the direction of heat current. This gives us further insight into the underlying mechanism.
5. Takeda *et al.* attributed the planar K_{xy} to topological magnons, while our results show strong evidence that the thermal Hall effect in Na₂Co₂TeO₆ is due to phonons.

So far, a planar thermal Hall effect has only been attributed to Majorana fermions [8] or topological magnons [3,6,7]. Our study provides the **very first** example that phonons are also able to generate a planar thermal Hall effect.

Note that the fundamental mechanism responsible for the phonon thermal Hall effect is still unsolved, even in the conventional configuration (with an out-of-plane magnetic field perpendicular to the heat current), despite several recent experimental [32,34,35,41,42] and theoretical studies [37,A,B,C]. Our results reveal a new facet of the phonon thermal Hall effect, namely that it can also occur in a “planar” configuration. We expect our present paper to play a pivotal role in the theoretical understanding of the surprising and still baffling Hall effect of phonons, by reporting entirely new information.

Reviewer #3

Recently, thermal Hall effect of quantum magnets has received extensive research interests. An exciting experimental finding is the half-quantized thermal Hall conductivity K_{xy} in the Kitaev antiferromagnet α -RuCl₃ for an in-plane field in excess of 7 T, which was interpreted as evidence of itinerant Majorana fermions. The half-quantized K_{xy} plateau appears even for a planar thermal Hall configuration, i.e., when the magnetic field is applied within the 2D plane and parallel to the heat current J ($J // H // a$, where a is the crystal direction perpendicular to the Ru-Ru bond, the so-called zigzag direction). However, this phenomenon has not been fully confirmed and the magnons and phonons were also proposed to contribute to the planar thermal Hall effect of this famous material.

In this work, Chen *et al.* studied the planar thermal Hall effect of another Kitaev material Na₂Co₂TeO₆. The main experimental findings include: (i) there are non-zero K_{xy} for either $J // H // a$ or $J // H // a^*$, which are proposed to be forbidden by the two-fold rotational symmetry along these directions; (ii) there is close similarity between the temperature and field dependence of the planar K_{xy} and those of phonon-dominated K_{xx} ; (iii) the planar K_{xy} strongly depends on the direction of the heat current relative to the magnetic field, i.e., $J // H$ or J perpendicular to H . The main conclusion is that phonons are responsible for the planar thermal Hall conductivity.

In general, this work displays some peculiar experimental results, like the non-zero planar thermal Hall conductivity forbidden by the two-fold rotational symmetry. The conclusion, if correct, would be very important for understanding the present topic on the origin of thermal Hall effect in the Kitaev system. The paper was well written and is suitable for Nature Communications. However, I have one comment to be considered by the authors.

We thank the Reviewer for recognizing the novelty and the important role that our work plays in the field of thermal Hall effect.

The authors showed the temperature dependence of K_{xy} and K_{xx} in a few different magnetic fields and made the comparison. The key experimental result the authors concluded is the close similarity in the temperature and field dependence between K_{xy} and K_{xx} . Here, the question is: what is the detailed magnetic field dependence of K_{xy} and K_{xx} ? Probably, no material can display similar magnetic field dependence of K_{xy} between K_{xx} because of their different origins. Nevertheless, without the comparison of $K_{xy}(H)$ and $K_{xx}(H)$ data, it is questionable to claim the similarity of magnetic field dependence between them.

This Reviewer would like to have seen more detailed field dependence of $K_{xy}(B)$ and $K_{xx}(B)$.

However, our measurement approach is to fix the field and sweep the temperature. So we do not have $K_{xx}(B)$ and $K_{xy}(B)$ but only $K_{xx}(T)$ and $K_{xy}(T)$ at different values of B .

Hess and coworkers have reported a large number of $K_{xx}(T)$ curves for NCTO [27], for $B = 0, 3, 5, 6, 7, 8, 9, 10, 11, 12, 13, 14$ and 15 T. We reproduce their data in Fig. A.

We see that nothing happens to K_{xx} up to 5 T, for both $B // a$ and $B // a^*$ (Fig. A, panels b and e). We observe the same in our own K_{xx} data (Figs. 2a and 2c).

Increasing the field to 10 T, we see that the 10 T curve lies on top of the 5 T curve for $B // a$ (Fig. A, panel e). We observe the same in our own K_{xx} data (Fig. 2a). However, for $B // a^*$, the 10 T curve clearly lies above the 5 T curve (Fig. A, panel b), and again we observe the same in our own K_{xx} data (Fig. 2c).

Increasing now to 15 T, we see a huge increase in K_{xx} for both field directions, in both sets of data.

So there is a detailed agreement between the extensive K_{xx} data of Hess and coworkers and our own data at selected fields. We see that there is no need for us to have more field values, as we have captured the three distinct and essential regimes: 1) 5 T for the regime of no effect; 2) 10 T for the regime on the border of the critical field; 3) 15 T for the regime of a huge peak.

Hong et al. have argued convincingly that the K_{xx} in NCTO is carried by phonons. We show in Fig. B that our K_{xy} data mimic the K_{xx} data in detail. The conclusion is that K_{xy} must be dominated by phonons as well.

In the revised manuscript, we include a more detailed discussion on the field dependence of both $K_{xy}(H)$ and $K_{xx}(H)$ data to strengthen the claim of a phonon scenario.

References not listed in our initial manuscript:

[A] Sun, Chen & Kivelson, *Phys. Rev. B* **106**, 144111 (2022).

[B] Mangeolle, Balents & Savary, *Phys. Rev. X* **12**, 041031 (2022).

[C] Flebus & MacDonald, *Phys. Rev. B* **105**, L220301 (2022).

[D] Hirokane *et al.*, *Phys. Rev. B* **99**, 134419 (2019).

List of changes to the manuscript:

- 1) Additional discussion: comparison of K_{xx} and K_{xy} to support the phonon scenario.
- 2) Additional discussion: T dependence of the ratio $|K_{xy} / K_{xx}|$.
- 3) Additional discussion: extrinsic scenario for Majorana fermions and magnons.
- 4) Additional discussion: relative directions of current and field.
- 5) Additional discussion: detailed field dependence of K_{xx} and K_{xy} .

REVIEWERS' COMMENTS

Reviewer #1 (Remarks to the Author):

The authors have addressed all the points I raised in my first report. Accordingly, they have amended the manuscript. I believe that the content of this paper is worthy of publication.

Reviewer #2 (Remarks to the Author):

I have reviewed the revised version of the manuscript and the author's response to the referees' comments. It appears that the authors have adequately addressed the comments and questions raised. I concur with their points about comparing them with the previous reports. However, I think this does not necessarily qualify the manuscript for publication in Nature Communications. As noted by the authors, the interpretation of thermal transport is subject to considerable debate, raising more questions than it answers. The experimental results presented here, although improved in terms of experimental error, magnitude, and the addition of new configuration, offer an alternative scenario for phonon contribution without providing more conclusive evidence. The comparison with the study of Takeda et al. highlights significant differences but ends with unresolved questions regarding the origin of the observed phenomena. Questions remain, such as whether the larger magnitude of the planar K_{xy} could exclude Takeda et al.'s interpretation and indeed support the phonon contribution or if the difference in magnitude between H parallel to J and H perpendicular to J reflects or supports their primary argument of the phonon contribution. Consequently, I am not convinced of the manuscript's suitability for publication in Nature Communications, given the lack of definitive evidence to support their claims.

Reviewer #3 (Remarks to the Author):

I am satisfied with the revised manuscript and the authors' response. I believe this work is very important for understanding the present topic on the origin of thermal Hall effect in the Kitaev system. Therefore, I recommend publishing this manuscript in Nature Communications.

Reviewer #1 (Remarks to the Author):

The authors have addressed all the points I raised in my first report. Accordingly, they have amended the manuscript. I believe that the content of this paper is worthy of publication.

We thank the reviewer for recommending publication.

Reviewer #2 (Remarks to the Author):

I have reviewed the revised version of the manuscript and the author's response to the referees' comments. It appears that the authors have adequately addressed the comments and questions raised. I concur with their points about comparing them with the previous reports. However, I think this does not necessarily qualify the manuscript for publication in Nature Communications. As noted by the authors, the interpretation of thermal transport is subject to considerable debate, raising more questions than it answers. The experimental results presented here, although improved in terms of experimental error, magnitude, and the addition of new configuration, offer an alternative scenario for phonon contribution without providing more conclusive evidence. The comparison with the study of Takeda et al. highlights significant differences but ends with unresolved questions regarding the origin of the observed phenomena. Questions remain, such as whether the larger magnitude of the planar K_{xy} could exclude Takeda et al.'s interpretation and indeed support the phonon contribution or if the difference in magnitude between H parallel to J and H perpendicular to J reflects or supports their primary argument of the phonon contribution. Consequently, I am not convinced of the manuscript's suitability for publication Nature Communications, given the lack of definitive evidence to support their claims.

We disagree with the reviewer's comment that our work does not resolve the question of what are the dominant heat carriers responsible for the thermal Hall effect in NCTO.

The first and major question that our work was aiming to answer is: could phonons also give a planar thermal Hall effect? The answer is YES. The detailed comparison between the phonon-dominated K_{xx} and K_{xy} provides compelling evidence that the excitations contributing to K_{xx} must also dominate the K_{xy} contribution. At 5 T, both K_{xy} and K_{xx} decrease monotonically to zero with decreasing temperature, whereas at 15 T, both K_{xy} and K_{xx} show a huge peak at low temperature. This is seen for both field directions ($B // a$ and $B // a^*$). At 10 T, there is a clear difference between $B // a$ and $B // a^*$: for $B // a$, both K_{xy} and K_{xx} show a similar monotonic decrease, as for the case of 5 T; for $B // a^*$, however, where the critical field is lower, 10 T is enough to cause the start of a rise at low T in K_{xx} , and, remarkably, this feature is also seen in K_{xy} . The fact that $K_{xy}(T,H)$ mimics the behavior of the phonon-dominated $K_{xx}(T,H)$ is compelling evidence that K_{xy} is also dominated by phonons. The comparison of our work with Takeda et al. provides an

alternative interpretation of the planar thermal Hall effect in $\text{Na}_2\text{Co}_2\text{TeO}_6$, in terms of phonons as the key heat carriers.

As we have said in the previous reply, the microscopic origin behind the phonon thermal Hall effect still remains an open question. Our findings of a strong dependence of K_{xy} on the current direction could shed new light on the phonon thermal Hall effect and strongly limit the theories that are compatible with such a scenario.

Reviewer #3 (Remarks to the Author):

I am satisfied with the revised manuscript and the authors' response. I believe this work is very important for understanding the present topic on the origin of thermal Hall effect in the Kitaev system. Therefore, I recommend publishing this manuscript in Nature Communications.

We thank the reviewer for recognizing the importance of our work and for recommending publication.